# Isolation and Characterization of Cell Envelope Fragments Comprising Archaeal S-Layer Proteins

**DOI:** 10.3390/nano12142502

**Published:** 2022-07-21

**Authors:** Kevin Pfeifer, Eva-Kathrin Ehmoser, Simon K.-M. R. Rittmann, Christa Schleper, Dietmar Pum, Uwe B. Sleytr, Bernhard Schuster

**Affiliations:** 1Institute of Synthetic Bioarchitectures, University of Natural Resources and Life Sciences, 1190 Vienna, Austria; kevinpfeifer@boku.ac.at (K.P.); eva.ehmoser@boku.ac.at (E.-K.E.); uwe.sleytr@boku.ac.at (U.B.S.); 2Archaea Physiology & Biotechnology Group, Department of Functional and Evolutionary Ecology, University of Vienna, 1030 Vienna, Austria; simon.rittmann@univie.ac.at; 3Arkeon GmbH, 3430 Tulln an der Donau, Austria; 4Archaea Biology and Ecogenomics Division, Department of Functional and Evolutionary Ecology, University of Vienna, 1030 Vienna, Austria; christa.schleper@univie.ac.at; 5Institute of Biophysics, University of Natural Resources and Life Sciences, 1190 Vienna, Austria; dietmar.pum@boku.ac.at

**Keywords:** archaea, S-layer protein, Archaea Biotechnology, molecular sieving, extraction method, biomimetics

## Abstract

The outermost component of cell envelopes of most bacteria and almost all archaea comprise a protein lattice, which is termed Surface (S-)layer. The S-layer lattice constitutes a highly porous structure with regularly arranged pores in the nm-range. Some archaea thrive in extreme milieus, thus producing highly stable S-layer protein lattices that aid in protecting the organisms. In the present study, fragments of the cell envelope from the hyperthermophilic acidophilic archaeon *Saccharolobus solfataricus* P2 (SSO) have been isolated by two different methods and characterized. The organization of the fragments and the molecular sieving properties have been elucidated by transmission electron microscopy and by determining the retention efficiency of proteins varying in size, respectively. The porosity of the archaeal S-layer fragments was determined to be 45%. S-layer fragments of SSO showed a retention efficiency of up to 100% for proteins having a molecular mass of ≥ 66 kDa. Moreover, the extraction costs for SSO fragments have been reduced by more than 80% compared to conventional methods, which makes the use of these archaeal S-layer material economically attractive.

## 1. Introduction

A proteinaceous array in bacterial cell envelope fragments was first suggested in the 1950s [1,2] and, approx. 15 years later, evidence could be provided that a protein layer is located as the outermost structure on the cell envelope surface of intact bacteria [3,4]. This so-called Surface(S)-layer can be found in most bacteria and almost all archaea [5,6,7]. S-layer proteins and glycoproteins (SLPs) form a crystalline and isoporous lattice, which covers the entirety of the cell surface [8,9]. The functions of the S-layer lattice are still not fully elucidated. *Deinococcus radiodurans*, a radiation-resistant bacterium, is one of the most studied model organisms for understanding the S-layers’ architecture and its function [10,11,12,13]. Functional characterization of the S-layer of *D. radiodurans* confirmed its transport properties; the nonselective permeation of ions [12,14] showed activity with different types of amino acids, behaving as a sieve barrier able to protect while allowing exchange and communication with the environment [13]. Further studies on various organisms propose that the S-layer may act as an ion trap, molecular sieve, cell-shape determining component, and it has been shown to be involved in cell surface recognition and cell division [6,15,16,17,18]. Because SLPs as the outermost cell envelope component are in direct contact with the environment, it has been proposed that the SLPs have evolved to contribute to the protection of the cells from the natural environments. This function is most conceivable for hyperthermophilic, acidophilic, and halophilic microorganisms [15,16,19]. Studies on SLPs across bacterial and archaeal kingdoms have shown that, although the protein sequences as well as tertiary and quaternary structures differ vastly among all studied samples, the lattices formed by the proteins always showed one of three basic symmetries, which are oblique (p1, p2), square(p4) or hexagonal (p3, p6). The unit cell dimensions of these geometries range from 3 to 30 nm with uniform pores in the range of 2 to 8 nm creating a structure with a porosity between 30 and 70% [6,20,21]. It has been shown that, under the proper conditions, purified bacterial SLPs can spontaneously reassemble into their native lattice geometry in solution, on solid supports, and on interphases like lipid films [6,22,23,24]. These properties highlight the vast potential of SLPs as building blocks in biobased materials with various applications such as biosensors, nanocarriers for drug/gene delivery, immobilization matrices, adjuvants, matrices for controlled biomineralization and as ultrafiltration membranes [6,25,26,27,28].

The present study focused on the S-layer large protein (SlaA) from *Saccharolobus solfataricus* P2(SSO). SSO has its optimum growth conditions at 78 °C and pH 2.5. The cytoplasmic membrane is composed of (di)ether and (tetra)ether lipids, which allow archaea to adapt to harsh conditions [29,30]. The structure of the SSO S-layer has recently been studied at a 16 Å resolution and is composed of two SLPs: the surface exposed SlaA and transmembrane S-layer small protein SlaB [31]. SlaA and SlaB protein occur in a ratio of 2:1, with each unit cell containing three SlaA dimers and three SlaB monomers [7,26,31]. The SlaB stalk anchors the SlaA proteins to the cell surface creating a pseudo-periplasmatic space of 25 nm in height, while the SlaA proteins organize into a lattice with a p3 symmetry featuring circular pores of ~4.5 and ~8 nm in diameter [31]. The high glycosylation of both SlaA and SlaB adds a layer of protective sugar coating, which may modulate the antifouling characteristics of the outermost cell envelope structure [32]. The native stability of these proteins against both extreme heat and acidic conditions, as well as the ease of cultivation of the strain, make SSO SLPs interesting building blocks for biotechnological applications.

SLPs from most organisms can be extracted by chemical or physical lysing of the cells and removing the released cytosolic components. The composition of the extracted cell envelope (type of underlying polymer (if any), Gram-positive or Gram-negative envelope structure, interaction with cytoplasmic membrane, chemistry of lipids, etc.), and thus the steps needed to purify the SLPs depend strongly on the respective organism. Most often hydrogen-bond breaking agents such as guanidine hydrochloride and urea or chelating agents such as EDTA are used to disassemble the extracted S-layer lattices and purify the SLPs. Once purified, it has been shown that the bacterial SLPs can spontaneously reassemble into their native lattice structure under proper conditions [6,22,23,24].

In the case of SSO, the extraction protocols found in recent publications are based on protocols developed for *Sulfolobus acidocaldarius* by Grogan and Michael et al. in the 1980s [33,34]. These protocols use detergents to chemically lyse the SSO cells and consequently remove the cytosolic fraction leaving behind a ghost cell, which is an empty shell of ether lipids and membrane associated proteins. In the case of SSO and other Sulfolobales, the SLPs are highly stable against various detergents; therefore, SLPs can be enriched from the ghosts by repeatedly washing them with detergents resulting in SlaA-enriched sacculi in the native cell shape (Appendix A) [34,35,36]. Although recent improvements have been proposed for the handling of the samples during the extraction process, the protocols have not been evaluated nor optimized for industrial applications.

## 2. Materials and Methods

### 2.1. Cultivation

*Saccharolobus solfataricus* P2 (SSO) was cultivated in 100 mL (50 mL *wt**/vol*) or 1 L (500 mL *wt**/vol*) flasks under shaking conditions at 78 °C, pH 3, in 50 mL Brock T/S medium supplemented with 0.1% (*wt/vol*) tryptone (T) and 0.2% (*wt/vol*) (+)D-Sucrose (S). Appendix A specifies the cost for the cultivation of SSO.

### 2.2. Extraction of SSO Ghosts and Production of SSO Sacculi and Fragments

SSO ghosts were isolated by the extraction method described by Gambelli et al. [31] using DNase (DNase-based Extraction of Ghosts; DEG) and by a modified extraction method as described below (Sonication-based Extraction of Ghosts; SEG). The sequential steps and terminologies throughout the SLP extraction process is shown in Appendix A. Cells were harvested from a 50 mL exponential growing culture by centrifugation at 2000× *g* for 30 min. The cells were resuspended in 10 mL Buffer A (10 mmol L^−1^ NaCl, 0.5% sodium lauroylsarcosine) and incubated at 37 °C and 80 rpm for 1 h. This buffer used in SEG differs from the DEG protocol in that no phenylmethylsulfonyl fluoride (PMSF; 1 mmol L^−1^) was added, and DNase I (10 µg mL^−1^) digestion of DNA was omitted. Instead, the suspension was sonicated for 30 s with 10 A (Thermo Fischer Q700, Thermo Fisher Scientific Inc., Eindhoven, The Netherlands) to shear the DNA. Subsequently, the SSO ghosts were pelleted by centrifugation at 20,000× *g* for 30 min. The resultant pellet was resuspended in 10 mL buffer B (10 mmol L^−1^ NaCl, 0.5 mmol L^−1^ MgSO_4_, 0.5% sodium dodecylsulfate (SDS)), incubated at 40 °C for 1 h and then pelleted by centrifugation at 20,000× *g* for 30 min. This process was repeated 3 times to completely remove lipids and SlaB. Subsequently, the obtained SSO sacculi comprising SlaA (UniProt KB entry: Q980C7) were washed twice with Milli-Q water (MQ). Protein concentration of the SSO sacculi was determined by spectrophotometric measurements (280 nm; ε(×1000): 177.6; M_r_ 131 kDa). The sacculi produced by the SEG method have been used for all further experiments. An aliquot of SSO sacculi equating to 300 µg protein was diluted to 2 mL with MQ and sonicated with 10 A for 15 s to gently break up the SSO sacculi to form large SSO fragments (Appendix A).

### 2.3. SDS Polyacrylamide Gel Electrophoresis (PAGE)

For all protein gels shown, samples were denatured in Laemmli Buffer at 98 °C for 20 min and run on a 4–12% Bis-Tris Gel (Invitrogen, NuPAGE Bis-Tris, Carlsbad, CA, USA) in MOPS buffer at 140 V until the smallest marker had reached the bottom of the gel. The gels were stained in a methanol free Coomassie-G250 solution by microwaving the stain covered gel for 45 s and subsequent shaking of the gel for 30 min. The gel was de-stained in MQ, imaged (Samsung, SM-G950F, Vienna, Austria), and processed in Photoshop.

### 2.4. Electron Microscopy

For transmission electron microscopy (TEM), 10 µL samples were incubated on a glow discharged grid (CF100H-Cu, EMS) for 2 min and stained with 2% uranyl acetate for 30 s. All samples were analyzed using a Libra 120 microscope (Zeiss, Wetzlar, Germany). Fourier domain processing was used to enhance the signal-to-noise ratio in the TEM images. The image processing routines were developed in-house as plugins for the open-source software ImageJ [37]. The central part of the Fourier spectrum of a TEM image of an SSO fragment is shown in Appendix A.

Samples were characterized by a scanning electron microscope (SEM) with a Thermo Fisher Scientific Apreo VS SEM (Thermo Fisher Scientific Inc., Eindhoven, The Netherlands). For this purpose, the filter membranes were cut into approx. 7 × 7 mm sized pieces and fixed with a conductive double sticky tape on standard 0.5” aluminum stubs. The SEM was operated at 5.0 kV with a beam current of 0.1 nA in standard mode. Images were recorded in high vacuum with the standard electron detector.

### 2.5. Deposition of SSO Fragments on Microfilters

The 2 mL SSO fragment suspension containing 300 µg protein (this equals 60 µg protein per cm^²^ microfilter area) was filled up to 12 mL MQ. An Amicon^®^ cell equipped with magnet stirrer and a volume of 12 mL was assembled with a microfilter clamped by an O-ring to the bottom the cell. Two different microfilters that were 2.5 cm in diameter and having a pore size 0.2 µm were used: a track-etched hydrophilic polycarbonate filter (MF1; Millipore, Darmstadt, Germany, GTTP02500) and a foam-like cellulose acetate filter (MF2; Sartorius AG, Göttingen, Germany, 11107-25-N) (see Appendix A) [38]. The microfilters were washed with MQ prior to use. The suspended SSO fragments were transferred into the Amicon^®^ cell. In all washing steps and filtration experiments, a pressure of 2 bar was applied to the Amicon^®^ cell and the stirring speed was 200 rpm at room temperature (22 °C). The SSO fragments collected on MF1 or MF2 were chemically cross-linked with 5 mL 2.5% glutaraldehyde for 20 min. The cross-linked SSO fragments on MF1 or MF2, further referred to as SSOMF1 and SSOMF2, respectively were washed twice by running 10 mL MQ through the composite structure. The integrity of SSOMF1 and SSOMF2 was checked by filtration of a solution of ferritin (Sigma, Darmstadt, Germany: F4503; 0.7 g L^−1^) as described previously through spectrophotometry [38]. Ferritin, a large globular protein with a molecular mass of 474 kDa and a size of 12 nm, exceeds the size of the pores in the S-layer lattice at least by a factor of 1.5. Therefore, ferritin must be completely rejected by an intact layer of SSO fragments on the microfilters.

### 2.6. Investigation of Retention Efficiency against Proteins by SSO Fragments

To test the filter characteristics of the pores within the SSO fragments, 4 mL of 0.7 g L^−1^ myoglobin (Sigma Aldrich, Darmstadt, Germany: M1882), carbonic anhydrase (Sigma Aldrich: C2624), horseradish peroxidase (Sigma Aldrich: 77332), bovine serum albumin (Sigma Aldrich: A2153), amyloglucosidase (Sigma Aldrich: 10115), and γ-globulin (Sigma Aldrich: G7516) were run through the SSO fragments in different order, i.e., from low molecular mass (M_r)_ to high M_r_ and *vice versa*. The retention efficiency of the deposited and cross-linked SSO fragments was determined by collecting the permeate after discarding the first milliliter. The measurements were performed in the Amicon^®^ cell at 2 bar and a stirring speed of 200 rpm. The protein concentration of the feed (initial concentration) and permeate were measured using a Nanodrop spectrophotometer (ND-1000, PEQLAB Biotechnologie GmbH, Erlangen, Germany). The retention efficiency was calculated by:$$\text{Retention}\;\text{efficiency}\;\left[\%\right]=\left(1-\frac{\text{Protein}\;\text{Concentration}\;\text{Permeate}\;}{\text{Protein}\;\text{Concentration}\;\text{Feed}}\right)\times 100$$

Between the runs, SSOMF1 and SSOMF2 were washed in the same orientation; then, filtration was performed with 10 mL MQ. Integrity and level of obstruction were inferred from the change in flux by measuring the time that it took to run 5 mL MQ through SSOMF1 and SSOMF2 with an active filtration area of 4.5 cm^2^. The concentration of all protein solutions used for filtration experiments was determined through spectrophotometry at 280 nm before (feed) and after (permeate) filtration.

## 3. Results and Discussion

### 3.1. A Modified and Cost-Effective Extraction Method for SSO Ghosts and Production of SSO Sacculi Comprising SlaA

Two different protocols to prepare SSO sacculi comprising SlaA from extracted SSO ghosts were conducted and evaluated for their effectiveness and costs. The established DEG method by Gambelli et al. [31] relies on DNase I for the digestion of the DNA, which is released during the chemical lysis of the cells. When calculating the extraction costs for SSO ghosts and subsequent sacculi production, it turned out that DNase I accounts for 58% of the material costs (Appendix A). To reduce the extraction costs, DNase I was eliminated from the extraction protocol, and the DNA released during cell lysis was instead sheared through sonication. Furthermore, costs and handling efforts were reduced by removing the protease inhibitor PMSF from the extraction buffer and reducing the extraction volume from 40 mL to 10 mL per 50 mL culture. TEM images of the prepared ghosts showed that those resulting from the DEG method were largely intact and the S-layer appeared to retain the cell shape (Figure 1A). The ghosts prepared through the SEG method, on the other hand, appeared slightly fragmented. The SEG method was refined to minimize damage to the ghosts, while sufficiently shearing the DNA to allow for processing (Figure 1B,C).

To compare the efficiency of the SEG to the DEG method, ghosts were prepared from three 50 mL cultures in the exponential growth phase with each extraction method. SDS-PAGE analysis of the extracts showed that both methods were successful in extracting ghosts and subsequent production of sacculi comprising SlaA by removing the intracellular components, lipids, and the membrane-hooked SlaB. While SEG samples showed reduced protein concentration compared to DEG samples, both extraction methods produced sacculi highly enriched in SlaA, which showed no noticeable impurities when analyzed using SDS-PAGE (Figure 2). The protein concentration of the purified SlaA proteins was determined spectrophotometrically resulting in a 21% decrease in protein concentration in the SEG samples (0.96 mg ± 0.06 mg) compared to the DEG samples (1.21 mg ± 0.08 mg). Although the yield of the SEG extraction method was lower, the costs per mg of SlaA extracted by sonication (0.17 ± 0.01 € mg^−1^; *n* = 3) were 81% lower than those calculated for the DEG extraction method (0.84 ± 0.05 € mg^−1^; *n* = 3) (Appendix A). Furthermore, SEG with identical parameters was applied for the extraction of ghosts from 500 mL of culture, yielding ~10 mg enriched SlaA after processing. Therefore, the SEG method is a promising and scalable alternative to the established extraction methods and, by bringing down production costs, advances the biotechnological potential of archaeal S-layers.

### 3.2. Characterization of SSO Fragments on Microfilters

Having established a cost-efficient extraction method, the SSO fragments produced by the SEG method were investigated for their physicochemical parameters, filtration properties and thus potential biotechnological applications. For the construction of SSOMF1 and SSOMF2, ghosts comprising SlaA were extracted and purified from a 500 mL exponentially growing SSO culture with a yield of 10 mg. After sonication, TEM images revealed broken up SSO sacculi fragments ranging from 200 to >3000 nm in size (Figure 3A). The sonication step had to be optimized to obtain large fragments because both intact or partly fragmented sacculi and too small SSO fragments did not form a closed and coherent layer on the microfilters. Nevertheless, TEM investigations confirmed for large SSO fragments the two types of pores as previously described (Figure 3B) [31] and showed for the first time that the hexagonal p3 lattice of the SSO fragments exhibits a porosity of up to 45% (Appendix A). Moreover, it could be confirmed that the SSO fragment constitutes a crystalline patch formed by SlaA (Figure 3B).

This suspension of large SSO fragments was diluted, transferred onto M1 and M2 (Appendix A), forced to sediment by applying pressure, and finally chemically cross-linked. SEM images of SSOMF1 at a magnification of 100,000× showed that MF1 was completely covered by SSO fragments (Figure 3C). The integrity of SSOMF1 and SSOMF2 was checked by filtration of a ferritin solution. The retention efficiency of SSOMF2 was determined to be 100%, indicating that the SSO fragments could form a dense layer on MF2 due to its foam-like structure (Appendix A). For SSOMF1, however, not the entire ferritin could be rejected, indicating that the structure of MF1 did not allow the SSO fragments to be pushed into the microfilter structure and to form a totally dense filtration layer. With bacterial S-layer material, both types of microfilter could be tightly covered [25,38,39,40,41], but, for biotechnological utilization of these filter membranes, it turned out that open-celled foam-like microfilter membranes like MF2 gave better and more reproducible filtration membranes [38,41].

Plain MF1 and MF2 (Appendix A) evinced a water (MQ) flux of 8800 L h^−1^ m^−2^ and 12,100 L h^−1^ m^−2^, respectively. The water flux of SSOMF1 and SSOMF2 was determined to be 32 L h^−1^ m^−2^ and 60 L h^−1^ m^−2^, respectively. In comparison, a higher flux was determined for bacterial S-layer material (i.e., cell wall fragments and/or peptidoglycan-containing sacculi) deposited on microfilters. In place, the flux ranged from 150 L h^−1^ m^−2^ to 250 L h^−1^ m^−2^, when measured at a pressure of 2 bar with MQ through a 0.22 µm microfilter membrane. In general, the water flux strongly depends on the amount of S-layer protein deposited per membrane unit area as a noticeable flux decline is observed with increasing amounts of S-layer material. Interestingly, the latter showed no influencing effect on the rejection characteristics [38]. In turn, the S-layer material required for the formation of defects free filtration layer is determined by porosity, the pore size, and the pore size distribution of the supporting microfilter membrane [38,39,42]. The microfilters used with the bacterial S-layer material and archaeal ones are different and hence allow *per se* no conclusive comparison of the flux. The latter strongly depends on the nature of the supporting membrane as well as on the amount of S-layer material deposited [38]. A linear relationship between the flux and amount of S-layer material used per membrane unit area has been observed [25,40]. In the present study, 60 µg SLP per cm^²^ microfilter area was used, whereas, in the bacterial system, in most cases, only half the amount of S-layer material was applied. This issue might be optimized so that a lower amount of archaeal S-layer material will be sufficient to obtain a dense and defect free filtration layer. However, this was not in the scope of the present study as the change of flux is not associated with alterations in retention efficiency of the SLP fragments. Another reason why the flux in the bacterial system is higher might be the fact that, in most cases, peptidoglycan-containing sacculi were deposited on microfilters, where peptidoglycan showed a much higher “pore” size and acted somehow as a spacer. In contrast, archaeal SSO fragments is comprised almost exclusively of the protein SlaA and thus may form a denser layer on microfilters.

To investigate the molecular sieving characteristics of the pores in SSO fragments, proteins varying in size were passed through the pores of the SSO fragments with pressure as a driving force (Table 1). SSOMF1 and SSOMF2 that remained intact after filtration were rinsed with MQ and reused for the next protein. To ensure that the retention efficiency of sequentially tested proteins was not affected, MQ flux was measured after rinsing, and protein solutions were passed through in various consecutive sequences. As shown in Table 1, Figure 4 and Figure 5, and Appendix A, both SSOMF1 and SSOMF2 showed an increased retention efficiency for proteins with increasing M_r_. This trend was observed no matter in which consecutive sequence the proteins (from small to large M_r_ or from large to small M_r_) were passed through the SSOMF filtration layer and when a mixture of proteins was filtered (Figure 4B; Appendix A).

Figure 4B shows the results of filtration experiments, where the proteins listed in Table 1 were passed through the same SSOMF1 in consecutive order either from smallest to largest or vice versa with the SSOMF1 being washed with MQ between each measurement. Starting with proteins from small to large M_r_ resulted at first sight in a higher retention efficiency compared to the passage of the proteins in reverse order (from large to small), but only for the proteins having a M_r_ ≥ 44 kDa (Figure 4B). However, the data points of both consecutive sequences for protein passage are in between the standard deviation of each other. For myoglobin and carbonic anhydrase, almost identical data have been observed (Figure 4B). Thus, the triplicate data show that there is no significant difference in retention efficiency no matter in which order the proteins were passed through the pores of the SSOMF1 membrane (Figure 4). This allows for the conclusion that the proteins retained on the SSOMF1 were removed from the surface and pores of the latter by the washing step. Interestingly, studies on the S-layer lattice of various bacterial organisms showed diminished or even eliminated unspecific adsorption events and thus provided an efficient antifouling coating [32,43]. Hence, one may conjecture that the deposited SSO fragments may provide an antifouling coating as well.

A comparison of the data obtained for SSOMF2 with those from so-called S-layer ultrafiltration membranes (SUMs) comprising bacterial S-layer material revealed differences in terms of retention efficiency. The limiting pore diameter of *Bacillus stearothermophilus* strains and of *Clostridium thermohydrosulfuricum* L111-69 was reported to be 4.5 nm and 4–5 nm, respectively [42,44]. This pore diameter is in the same dimension as the smaller pore of ~4.5 nm in the S-layer lattice of SSO [31]. The retention efficiency differs from that of *B. stearothermophilus* based SUMs, which showed a low retention efficiency for carbonic anhydrase (M_r_ of 31 kDa) of only up to 5%. In contrast, SUMs composed of the S-layer lattice from *C. thermohydrosulfuricum*, but also of the presently used SSO fragments on MF2 revealed a retention efficiency between 60% and 66% (Figure 5) [25,39]. The divergence between the retention efficiency for the protein having a M_r_ of about 44 kDa may be attributed to the use of horseradish peroxidase (44 kDa) in the present study and ovalbumin (43 kDa) in the studies on the bacterial S-layer material [25,39].

The shape of the retention curve is similar for SUMs made from *C. thermohydrosulfuricum* and SSO fragments. SSO fragments, however, show two distinct types of pores (Figure 3B): a smaller one, which appears triangular shaped with a size of approximately 4.5 nm, and a larger pore with an almost circular shape with approximately 8 nm in diameter. From Figure 3B, one can estimate the distribution of the pores of two small to one large pore. One may assume that these larger pores may cause a lower retention efficiency of SSO fragments for horseradish peroxidase, bovine serum albumin and amyloglucosidase, because these proteins show a smaller molecular size (Table 1) and thus can theoretically pass the large pores. We did not find such a behavior indicating that the larger pores of the SSO fragments are not accessible to a large extent. A possible reason might be that the SSO fragments are deposited in a way that a predominant number of large pores are not accessible due to superposition of fragments. Finally, no sharp exclusion limit as it has been shown for *B. stearothermophilus* SLP based SUMs can be reported for SSO fragments. However, the pore of the S-layer lattice of *B. stearothermophilus* has a very similar pore size as those of *C. thermohydrosulfuricum*, but the retention efficiency was completely different for carbonic anhydrase, a protein with small molecular mass (31 kDa; Figure 5). This provided evidence that other factors like net charges and hydrophilic or hydrophobic surface properties may play a significant role for the sieving characteristics [45] and thus, influence the physiological functions of the S-layer lattice.

## 4. Conclusions and Outlook

This study showed the ability of the SEG method to reduce the cost for SLP extraction from *S. solfataricus* as isolated SSO fragments by approx. 80% when compared to the established extraction method (DEG). The characterization of SSO fragments by TEM revealed an SSO S-layer lattice geometry with two types of pores (4.5 nm and 8 nm in diameter), a unit cell size of the p3 lattice of 21.3 nm, and a porosity of the lattice was calculated to 45%.

To characterize the pores of the SSO lattice, SSO fragments were deposited on different microfilters and finally cross-linked. The type of structure of the microfilter has crucial influence regarding whether a closed coherent SSO filter layer can be formed or not. An open-foam-like structure was compared to the structure of the radiation-track microfilter more appropriate as the SSO fragments could better be pushed into the microfilter structure and thus formed a totally dense filtration layer. The second important parameter is the size of the SSO fragments. Only large fragments gave a closed SSO layer on the MF2, whereas SSO sacculi and small SSO fragments formed leaky layers on both microfilters.

The molecular sieving characteristics of the pores in SSO fragments as determined by passing proteins varying in mass and thus size through the pores of the SSO fragments revealed a higher retention efficiency with increasing molecular mass of the proteins (Table 1). A comparison of the retention efficiency of SSO fragments bacterial S-layer material from *C. thermohydrosulfuricum* deposited in a dense layer on microfilters showed similar sieving properties. This result cannot be explained by the pore size because the SSO fragments have two pores (4.5 nm and 8 nm in diameter) and those of *C. thermohydrosulfuricum* have only one pore with a diameter of 4–5 nm.

Studying the molecular sieving characteristics of SSO fragments may help to advance the fundamental understanding of archaeal SLP biochemistry and biophysics and would simultaneously expand their biotechnological relevance. Lastly, established genetic systems for SSO may allow for the modification of S-layer pore properties and the construction of recombinant SLPs to produce SSOMFs with tailored functions, particularly in the pore region. Moreover, the effect of the glycosylation on the filter function of the SSO lattice could be investigated in detail by comparing wild-type and recombinant SSO SLPs.

While all application-driven S-layer publications to date have focused on bacterial S-layer proteins [24,46], a recent review on the biotechnical potential of archaea showed that the production of archaeal SLP-based applications is currently at Bio-Technology Readiness Level 2 [26]. This indicates that archaeal SLPs could be an interesting extension to the arsenal of building blocks for synthetic bioarchitectures. One expected advantage when using archaeal SLPs is the increased stability of the latter against elevated temperature, strong acidic environment, or higher ion concentration as the SLPs as outermost cell envelope structure of archaea retain by nature their structure in contact with these extreme environmental conditions. Therefore, SSO fragments and other archaeal SLP material might have application potential analogous to the bacterial ones as immobilization matrix for bioactive molecules [47] and as support for functionalized lipid membranes [48]. In summary, we have also provided evidence for the potential of archaeal SSO fragments as filtration matrix and thus expanded the field of Archaea Biotechnology.

## Figures and Tables

**Figure 1 nanomaterials-12-02502-f001:**
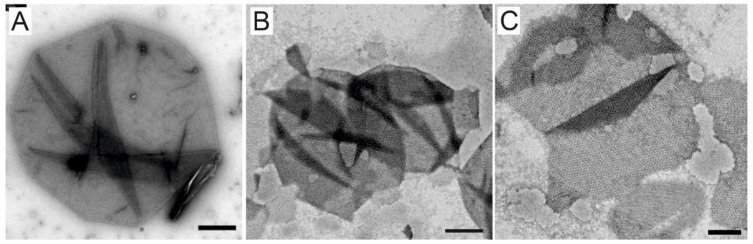
(**A**) *Saccharolobus solfataricus* P2 (SSO) ghost prepared using the extraction with DNase I (DEG). (**B**,**C**) SSO ghosts prepared using the extraction method with sonication (SEG). Scale Bars: A = 100 nm; B = 500 nm; C = 250 nm.

**Figure 2 nanomaterials-12-02502-f002:**
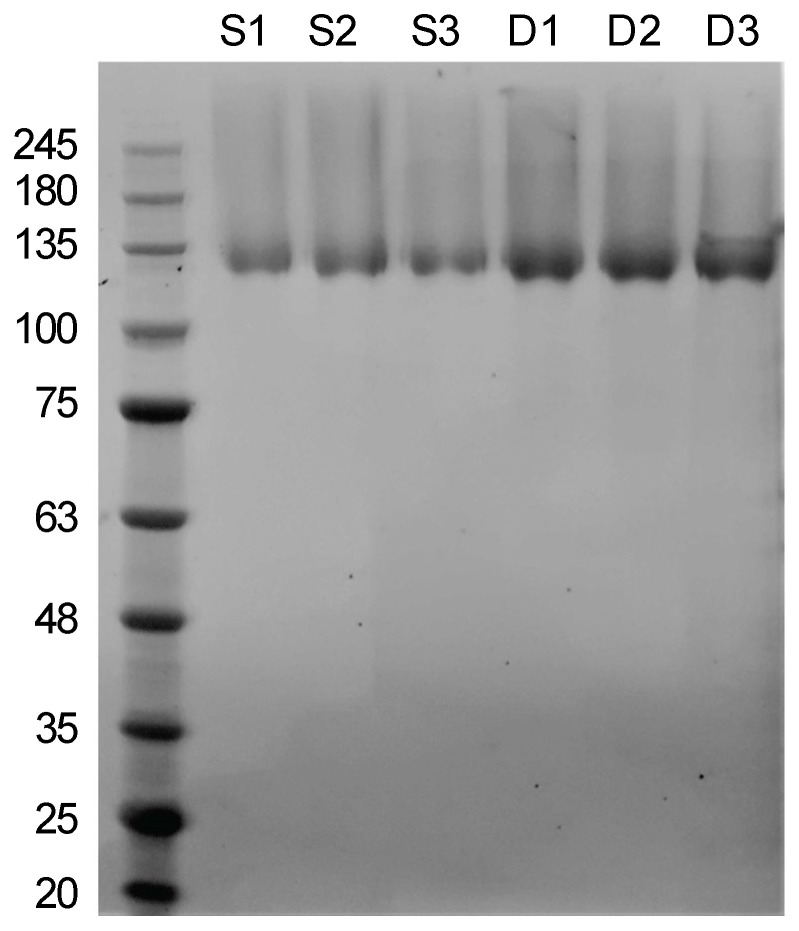
SDS-PAGE (4–20% Bis-Tris, MES) S1–S3: 20 µL of *Saccharolobus solfataricus* P2 (SSO) sacculi using the extraction method with sonication (SEG). D1–D3: 20 µL of *SSO* sacculi using the established extraction method with DNase I (DEG). The first lane corresponds to molecular mass standard with ladder units in kDa.

**Figure 3 nanomaterials-12-02502-f003:**
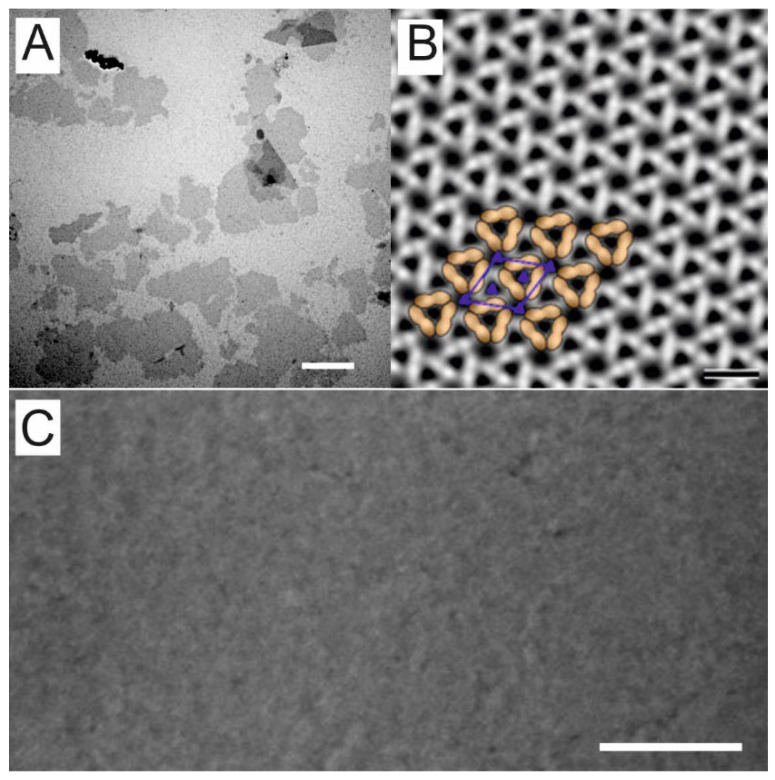
(**A**) Transmission electron micrograph of *Saccharolobus solfataricus* P2 (SSO) fragments after sonication. Bar, 1000 nm; (**B**) computer image reconstruction (2D projection) of the p3-ordered hexagonal S-layer lattice from SSO. Shown are a unit cell and the corresponding three-fold axis of rotation (symmetry operators) in blue. The protein is light, the pores are dark. In addition, a possible configuration of proteins belonging to a unit cell is shown by strokes drawn by hand in orange. Bar, 20 nm; (**C**) scanning electron micrograph of SSO fragments deposited on microfilter 1 (100,000× magnification). Bar, 500 nm.

**Figure 4 nanomaterials-12-02502-f004:**
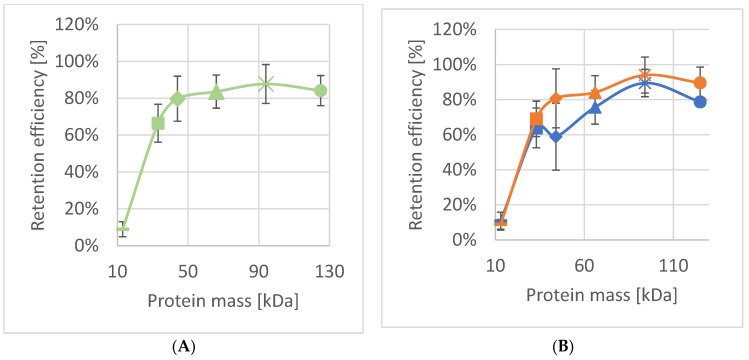
(**A**) Average efficiency of *Saccharolobus solfataricus* P2 fragments deposited on microfilter 1 (SSOMF1) across all experiments (*n* ≥ 6); (**B**) retention efficiency of SSOMF1 where Orange = filtration sequence largest to smallest protein (*n* ≥ 3) and Blue = filtration sequence smallest to largest protein (*n* ≥ 3); Legend for A and B: ▬ = myoglobin (17 kDa); ■ = carbonic anhydrase (31 kDa); ♦ = horseradish peroxidase (44 kDa); ▲ = bovine serum albumin (66 kDa); **x** = amyloglucosidase (97 kDa) and ● = γ-globulin (125 kDa).

**Figure 5 nanomaterials-12-02502-f005:**
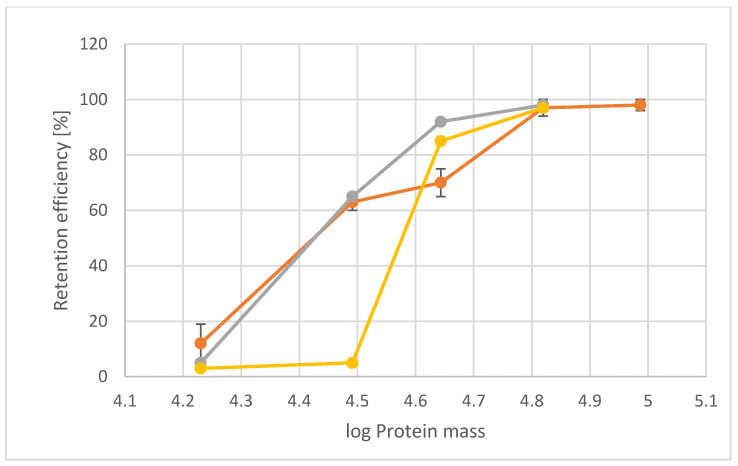
Retention efficiency of *Saccharolobus*
*solfataricus* P2 fragments deposited on microfilter 2 (SSOMF2) (orange); *B. stearothermophilus* pV72 (yellow) and *C. thermohydrosulfuricum* L111-69 (gray). The proteins with increasing molecular mass are myoglobin (17 kDa), carbonic anhydrase (31 kDa), ovalbumin (43 kDa; for bacterial SLP material), horseradish peroxidase (44 kDa; for SSOMF2), bovine serum albumin (66 kDa), and amyloglucosidase (97 kDa) (n ≥ 4). Data for the retention curve of *B. stearothermophilus* pV72 and *C. thermohydrosulfuricum* L111-69 are taken from Ref. [25].

**Table 1 nanomaterials-12-02502-t001:** Molecular mass (M_r_), molecular size, and retention efficiency of *Saccharolobus*
*solfataricus* P2 fragments deposited on microfilter 1 (SSOMF1) and 2 (SSOMF2), respectively, for selected proteins.

Protein	Molec. Mass M_r_ (kDa)	Molecular Size (nm)	Retention Efficiency SSOMF1	Retention Efficiency SSOMF2
Myoglobin	17	4.4 × 4.4 × 2.5	9% ± 4%; *n* = 9	12% ± 7%; *n* = 4
Carbonic anhydrase	31	4.1 × 4.1 × 4.7	66% ± 10%; *n* = 9	63% ± 3%; *n* = 4
Horseradish peroxidase	44	4.0 × 6.7 × 11.7	80% ± 12%; *n* = 6	70% ± 5%; *n* = 5
Bovine serum albumin	66	4.0 × 4.0 × 14.0	84% ± 9%; *n* = 7	97% ± 3%; *n* = 4
Amylo-glucosidase	97	5.7 × 7.3 × 10.7	88% ± 11%; *n* = 9	98% ± 2%; *n* = 4
γ-globulin	125	4.5 × 8.4 × 14.5	84% ± 8%; *n* = 6	

## Data Availability

The data presented in this study are available on request from the corresponding author.

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
