# Peer review of "Isolation and Characterization of Cell Envelope Fragments Comprising Archaeal S-Layer Proteins"

_nanomaterials, 2022, doi:10.3390/nano12142502_

Round 1
Reviewer 1 Report
The manuscript focuses on the interesting subject of exploiting S-layer proteins as molecular filters at nanoscale. The authors have a long and respected tradition on the field and on using these proteins as filters. In general, they present an alternative protocol for S-layer protein isolation and test it using two different types of filtering supports. Results and discussion are presented exhaustively but there are a few parts where they appear less clear (please see attached file for details). I find a bit confusing the use of sacculi and ghosts that sometimes seems to be interchanged. Acronym should be used more intuitively. Overall, I find the article very interesting.

Author Response
The manuscript focuses on the interesting subject of exploiting S-layer proteins as molecular filters at nanoscale. The authors have a long and respected tradition on the field and on using these proteins as filters. In general, they present an alternative protocol for S-layer protein isolation and test it using two different types of filtering supports. Results and discussion are presented exhaustively but there are a few parts where they appear less clear (please see attached file for details). I find a bit confusing the use of sacculi and ghosts that sometimes seems to be interchanged. Acronym should be used more intuitively. Overall, I find the article very interesting.
Answer: We have carefully checked and corrected the use of the terms “ghosts” and “sacculi”. Moreover, Figure S1 gives a clear overview about the terminology. All acronyms have been explained in the Abbreviation section.
The comments and concerns raised by reviewer 1 in the attached PDF-file are answered directly in this document and has been uploaded separately. Please see the attachment.

Reviewer 2 Report
The authors investigated two different methods for isolating and characterizing cell envelope fragments from the cell envelope of archaea consisting of a monomolecular protein lattice known as the surface layer.
The authors optimized both protocols studied and found that the price-performance ratio of the new SEG compared with the conventional DEG method. The results they reported can help scientists isolate components from various bacterial and archaeal cell walls.
The article contains numerous abbreviations, and it is very difficult to follow the text... Perhaps a glossary at the beginning of the article would be useful. For example, at the end of the article, I am still wondering what the abbreviations SEG and DEG mean.
Otherwise, the article is written correctly, and even though I am not from the field, I can follow the story by looking at the numerous abbreviations, which I assume are familiar to scientists in the field.
I recommend a minor revision - a glossary would help make the article understandable to a wider audience.
Author Response
The authors investigated two different methods for isolating and characterizing cell envelope fragments from the cell envelope of archaea consisting of a monomolecular protein lattice known as the surface layer.
The authors optimized both protocols studied and found that the price-performance ratio of the new SEG compared with the conventional DEG method. The results they reported can help scientists isolate components from various bacterial and archaeal cell walls.
The article contains numerous abbreviations, and it is very difficult to follow the text... Perhaps a glossary at the beginning of the article would be useful. For example, at the end of the article, I am still wondering what the abbreviations SEG and DEG mean.
Otherwise, the article is written correctly, and even though I am not from the field, I can follow the story by looking at the numerous abbreviations, which I assume are familiar to scientists in the field.
I recommend a minor revision - a glossary would help make the article understandable to a wider audience.
Answer: A glossary has been included in the Abbreviation section before the Reference section.
Reviewer 3 Report
· Throughout manuscript instead of using “sheer” and “sheered” for DNA removal use “shear” and “sheared”.
· Replace “SDS gel with “SDS-PAGE”
· Please described how Figure 3B was made, what software and parameters were used.
· in Figure 3C/D it is very hard to discern any data.
· Please describe how water flux determinations were made.
· For deposition of SSO fragments. Was the Amicon cell run to dryness before cross-linking?
· For retention efficiency, please clarify if this was done in the Amicon cell or in some other apparatus.
· For washes between filtration steps, please clarify if the filter was switched in orientation (backflushed) or just washed in the same orientation.
· Page 7: Instead of “halve” “half”.
· I find the data in Figure S4 to be more convincing than data shown in Figure 4, which basically seems to be a graphical representation of data shown in table 1. I suggest moving Figure 4 to supplemental data and including Figure S4 instead.
· Conclusion: Please clarify how scalability was shown or change this statement.
· Video was not made available for review.
Author Response
1) Throughout manuscript instead of using “sheer” and “sheered” for DNA removal use “shear” and “sheared”.
Answer: These typos have been corrected throughout the manuscript.
2) Replace “SDS gel with “SDS-PAGE”
Answer: Has been replaced accordingly
3) Please described how Figure 3B was made, what software and parameters were used. in Figure 3C/D it is very hard to discern any data.
Answer: The image processing routines were developed in-house as plugins for the open-source software ImageJ. Fig.3B was obtained by standard (“optical”) filtering. A digital version of the well-known optical mask used in diffractometers in the 1960’s and 1970’s was generated based on the reciprocal lattice which had been identified after indexing the peaks in the Fourier spectrum. Peaks were considered only if their intensity was higher than a threshold of 1.5 with respect to the average intensity of peaks in a surrounding area of 9x9 pixels. The diameter of the "punched" holes in the digital filter mask was set to 20% of the length of the basis vector. A cosine2-function was used to smooth the edge of the filter holes. The reconstructed image was finally obtained by applying the filter mask to the digital Fourier transform and performing the inverse Fourier transform. Information regarding this procedure has been added in the supplementary material (Figure S2 and figure legend).
Figure 3C/D has been replaced by a larger image with 100.000 magnification. As both images (Fig. 3C/D) showed the same SSOMF1, there was no need to show two images again.
4) Please describe how water flux determinations were made.
Answer: The water flux was determined by measuring the time which took to run 5 mL MQ through SSOMF1 and SSOMF2 with an active filtration area of 4.5 cm² (see last paragraph before section “3. Results and Discussion”).
5) For deposition of SSO fragments. Was the Amicon cell run to dryness before cross-linking?
Answer: Before crosslinking the water was squeezed out by applying a pressure of 2 bar. However, the Amicon cell was not completely dry, meaning that there is some water in the microfilter and in the SSO fragments.
6) For retention efficiency, please clarify if this was done in the Amicon cell or in some other apparatus.
Answer: All retention efficiency measurements were done in the Amicon cell. The use of the Amicon cell is also indicated in the Material and Method section starting from line 218.
7) For washes between filtration steps, please clarify if the filter was switched in orientation (backflushed) or just washed in the same orientation.
Answer: The filter was washed in the same orientation. It does not work to switch the orientation in the Amicon cell because it is not possible to place a disassembled SSOMFs in such a way that the O-ring in the Amicon cell will seal it completely again
8) Page 7: Instead of “halve” “half”.
Answer: Has been corrected
9) I find the data in Figure S4 to be more convincing than data shown in Figure 4, which basically seems to be a graphical representation of data shown in table 1. I suggest moving Figure 4 to supplemental data and including Figure S4 instead.
Answer: Figure 4 is important for the Discussion section as there is additional information presented, which is not given in Table 1. This is the case particularly for Figure 4B (proteins were passed through the same SSOMF1 in consecutive order either from smallest to largest or vice versa). Thus we prefer to keep this figure in the manuscript.
10) Conclusion: Please clarify how scalability was shown or change this statement.
Answer: This statement has been deleted.
11) Video was not made available for review.
Answer: We have not included any video neither in the manuscript nor in the supplementary material. Sorry for this misunderstanding.
Round 2
Reviewer 1 Report
The authors provided the changes/modifications/corrections as requested. Some minor issue still remain (please see the attached document).

Author Response
Correction as suggested by the reviewer have been made in line 47 and line 48. The entire manuscript has been checked for kDa and the numbers have been corrected accordingly. Typos have been found in line 75, 278, and 443, which have been corrected. References 23 and 27 are correct.
